# Fabrication and Investigation of PE-SiO_2_@PZS Composite Separator for Lithium-Ion Batteries

**DOI:** 10.3390/ma15144875

**Published:** 2022-07-13

**Authors:** Liguo Xu, Yanwu Chen, Peijiang Liu, Jianghua Zhan

**Affiliations:** 1College of Light Chemical Industry and Materials Engineering, Shunde Polytechnic, Foshan 528333, China; 21099@sdpt.edu.cn (L.X.); yanwuchen@126.com (Y.C.); 2Reliability Research and Analysis Centre, China Electronic Product Reliability and Environmental Testing Research Institute, Guangzhou 510610, China; 3China National Electric Apparatus Research Institute Co., Ltd., Guangzhou 510000, China

**Keywords:** composite membrane, SiO_2_@PZS nanoparticle, lithium-ion batteries

## Abstract

Commercial polyolefin separators exhibit problems including shrinkage under high temperatures and poor electrolyte wettability and uptake, resulting in low ionic conductivity and safety problems. In this work, core–shell silica-polyphosphazene nanoparticles (SiO_2_@PZS) with different PZS layer thicknesses were synthesized and coated onto both sides of polyethylene (PE) microporous membranes to prepare composite membranes. Compared to pure silica-coated membranes and PE membranes, the PE-SiO_2_@PZS composite membrane had higher ionic conductivity. With the increase in the SiO_2_@PZS shell thickness, the electrolyte uptake, ionic conductivity and discharge capacity gradually increased. The discharge capacity of the PE-SiO_2_@PZS composite membrane at 8 C rate was 129 mAh/g, which was higher than the values of 107 mAh/g for the PE-SiO_2_ composite membrane and 104 mAh/g for the PE membrane.

## 1. Introduction

Lithium-ion batteries (LIBs) have the advantages of high energy density, high output voltage, long cycle lives and low environmental pollution. Therefore, they not only play a significant role in the field of portable electronic products and electric vehicles but also a decisive one in promoting the adjustment of energy structures and ensuring the sustainable development of energy and the environment. LIBs have become the standard power supply for mobile electronic devices, such as mobile phones and laptops, and have begun to be applied to light electric vehicles and hybrid electric vehicles. Commercial LIBs have significant potential safety hazards due to the use of polyolefin separators with poor thermal stability and flammable and volatile carbonate solvents [1,2,3,4,5]. Furthermore, the lack of polar groups reduces the wettability and liquid absorption of the polyolefin separator and the existence of crystal structures also causes ion transport difficulties that reduce conductivity. Therefore, the performance of LIB separators and electrolytes is the decisive factor affecting the safety and functioning of LIBs.

Many researchers have investigated how to overcome those problems. One direction is the use of a solid polyelectrolyte (SPE) instead of a liquid electrolyte to avoid the leakage of organic solvents and the use of separators. Poly(ethylene oxide)-based (PEO) (co)polymer is the most widely studied SPE, but its room temperature conductivity is low, meaning that it cannot meet the requirements of practical applications [6,7,8,9,10,11,12]. Thus, modification of the separator (improvement of the wettability, electrolyte uptake, thermal stability, Gurley value, etc.) is another interesting method to solve these issues.

A variety of other methods have been developed in recent years to overcome the disadvantages of separators, including polymer composites (a blend of polymer and inorganic fillers) [13,14,15,16,17], inorganic fillers (e.g., SiO_2_, TiO_2_, Al_2_O_3_ and zeolite) [18,19,20,21,22,23], and polymer blends [17,24]. Remarkably, it has been confirmed that modification of a polyolefin membrane with a blend of polymer and inorganic fillers as a coating may be the most promising approach [1,25]. It not only maintains the microporous structure of the polyolefin separators but also improves the thermal stability, mechanical strength, wettability and ionic conductivity of the separators. Among the various inorganic fillers, SiO_2_ is the most widely studied due to its easy preparation, effective function and economic adaptability. A composite polyethylene (PE) separator was carefully prepared with a PEI/SiO_2_ blend and the electrolyte uptake, thermal stability, ionic conductivity and Li^+^ transference number were improved [26]. A core–shell structure with a SiO_2_@PMMA nanoparticle was successfully synthesized using PMMA as the shell layer, while the composite PE separator was fabricated with a SiO_2_@PMMA coating, which enhanced the thermostability, ionic conductivity, cycle performance and C-rate capability of the cells with the composite separator [27]. Recently, silica-poly (cyclotriphosphazene-*co*-4,4′-sulfonyldiphenol) core–shell nanoparticles (SiO_2_@PZS) were effectively synthesized and a novel composite PE-SiO_2_@PZS separator was obtained by coating the organic–inorganic nanoparticles on both sides of the PE membrane, and it demonstrated superior physical and electrochemical properties compared to the initial PE membrane and separator with the SiO_2_ coating [28]. The authors speculated that the surface hydroxyl groups and nitrogen and oxygen atoms from the outside layer of the PZS coordinated with the lithium ions, resulting in the improvement of the lithium dissociation, which further enhanced the electrochemical performance. Thus, the PZS shell played a pivotal role in the modification of the PE membrane. The thickness of the polymer shell was not studied. In addition, it has been reported that the particle size for the coating also has a decisive effect on the performance of the separator [29,30,31].

In the present work, SiO_2_ modified with a PZS outside layer—a core–shell structure of SiO_2_@PZS—was first prepared with different PZS layer thicknesses. Then, the organic–inorganic nanoparticle was fabricated to slurry with a polyvinyl alcohol (PVA) binder. Lastly, the slurry was coated onto both sides of a PE membrane to produce the composite separator (PE-SiO_2_@PZS). The effect of the SiO_2_@PZS size and PZS layer thickness on the electrolyte uptake, Gurley value and electrochemical performance of the separator was carefully investigated with a cell assembled using a LiCoO_2_ cathode and Li metal anode.

## 2. Materials and Methods

### 2.1. Materials

Hexachlorocyclotriphosphazene (HCCP) from Aladdin was twice purified through recrystallization from dry n-hexane before use. The PE membrane (thickness: 18 μm, porosity: 48%) and liquid electrolyte (LBC305-1) were purchased from Aldrich. Ammonia (25%), anhydrous ethanol, 4,4′-sulfonyldiphenol (BPS), tetraethyl orthosilicate (TEOS), acetonitrile (ACN), polyvinyl alcohol (PVA) and triethylamine (TEA) were used as received from Aladdin. Other materials from Yinuokai (China) and Aladdin were used as received.

### 2.2. Characterizations

Fourier-transform infrared spectroscopy (FT-IR) was performed with a Bruker Equinox 55 spectrometer with the range of 400–4000 cm^−1^. High-resolution TEM (HR-TEM; Philips CM200 FEG) was carried out to study the morphologies of the nanoparticles. A field emission scanning electron microscope (FE-SEM, SU8010, Hitachi) was used to observe the morphologies of the original PE membrane and the nanoparticle-coated composite membrane. The measurement of the electrolyte uptake (EU) was based on the weight change (EU% = (W2 − W1)/W1 × 100%; W1 is the original weight of the composite membrane and W2 is the weight of the separator soaked in the liquid electrolyte). A Gurley 4110 densimeter was used to test the air permeability of the membranes. The ionic conductivity (σ) of the membranes was measured using an AC impedance spectroscopic technique with a BioLogic Science Instrument VMP3B-10 workstation. The values of σ were calculated according to the equation: σ = L/(SR), where L is the thickness of the membranes, S is the area of the stainless-steel electrode and R is the bulk resistance of the membranes (marked as R_b_). The same workstation was used to measure the interfacial impedance of the membranes with a frequency range of 10 mHz–1 MHz (5 mV, 25 °C). The membranes were sandwiched between two stainless-steel electrodes in an SS/M/SS configuration and a liquid electrolyte mixture (volume ratio of EC/DEC/DMC = 1/1/1, with 1 M LiPF_6_) was added to the coin cell. A lithium metal anode and a LiCoO_2_ (Yinuokai, China) cathode were used to form a sandwich cell with the separators in the middle. The cells were processed through several discharges with current densities from 0.5 C to 8 C (2.75–4.2 V voltage) to test the C-rate capability. A LAND CT-4008 battery testing system was used to investigate the cycle stability of the cells (0.5/2 C, 100 cycles). X-ray diffraction tests were undertaken with an X’pert Pro Panalytical X-ray diffractometer at incident angles ranging from 5° to 70°.

### 2.3. Synthesis of SiO_2_ Nanoparticles

SiO_2_ nanoparticles were synthesized following a procedure from a previous publication [32]. The typical procedure was as follows: NH_3_·H_2_O (0.53 mol/L), ethanol and water were added to a flask (A) at room temperature with 30 min stirring. Then, ethanol and TEOS (0.53 mol/L) were added to another flask (B) to form a homogeneous solution. After that, the solution in B was quickly poured into A under stirring over 2 h. The reaction mixture was purified with centrifugation three times and ethanol was used to wash the nanoparticles. Finally, the nanoparticles were dried in a vacuum at 60 °C. The diameters of the prepared SiO_2_ nanoparticles were 281 ± 29 nm (Figure 1a).

### 2.4. Synthesis of Silica-Polyphosphazene Nanoparticles (SiO_2_@PZS)

SiO_2_@PZS nanoparticles were synthesized following a procedure from a previous publication [33]. The typical procedure was performed as follows: 200 mL ACN containing 1 g of prepared SiO_2_ was first stirred under ultrasonic conditions over 2 h, and then HCCP (0.004 mol), BPS (0.012 mol) and 8 mL TEA were quickly added to the solution. Subsequently, the reaction temperature was raised to 60 °C and the solution was continuously stirred for 8 h. Finally, the SiO_2_@PZS nanoparticles were purified with centrifugation three times and washed with water and acetone. The obtained white solid was dried in a vacuum at 80 °C for 24 h. SiO_2_@PZS nanoparticles prepared under other conditions used a similar process and the results are shown in Table 1. We regulated the thickness of the PZS mainly by controlling the concentrations of BPS and HCCP. When the concentrations of BPS and HCCP were too low, the thickness of the nanoparticles obtained was not uniform. When their concentrations were too high, the shell thickness of the obtained nanoparticles did not increase significantly. Hence, we chose a moderate concentration range.

### 2.5. Fabrication of PE-SiO_2_@PZS Composite Membrane

The typical procedure was performed as follows: PVA was used as the binder to enhance the adhesiveness of the nanoparticle slurry and membrane. First, SiO_2_@PZS (0.5 g) nanoparticles were added to 10 mL deionized water and 2 h of ultrasonication was deployed to ensure the effective l dispersion of the nanoparticles. Then, 0.1 g of PVA was added to the solution and the mixture was stirred for another 8 h. Finally, the obtained slurry was coated onto both sides of the PE membrane and the fabricated composite membrane (SiO_2_@PZS/PE) was then dried in a vacuum at 80 °C for 24 h. The same amount of slurry was applied to the PE surfaces for different groups. The composite membrane fabricated with pure SiO_2_ was prepared with a similar process and designated SiO_2_/PE.

## 3. Results and Discussion

### 3.1. Characterization of SiO_2_@PZS Nanoparticles

The uniformity of nanoparticles on the coating membrane in LIBs directly affects the conductivity and stability. Therefore, we first synthesized SiO_2_ nanoparticles with a uniform size in order to prepare coated nanoparticles with a uniform size and SiO_2_ nanoparticles as the core (Figure 1a). The SiO_2_@PZS nanoparticles were prepared following the method developed by Huang et al. [33]. The crosslinking reaction between BPS and HCCP takes places very easily under alkaline condition and the polymer PZS generated by the crosslinking reaction grows on the surface of SiO_2_, which was here considered as a hard template core in the coating reaction. Therefore, the thickness of the coating layer can be controlled by controlling the initial reactant feed ratio, reaction temperature and reaction time, which have a great effect on the absorption of the liquid electrolyte and even on the conductivity of LIBs.

FT-IR was used to investigate the coating reaction (Figure 2). The sharp peaks at 1490 cm^−1^ and 1590 cm^−1^ were due to the stretching vibration of the benzene ring skeleton of BPS and the typical peaks for O=S=O of BPS could also be observed at 1150 cm^−1^ and 1290 cm^−1^ [34,35]. The peaks at 885 cm^−1^ and 1190 cm^−1^ are attributable to the stretching vibration of the P=N double bond and P-N bond, which results from the cycle structure of HCCP. In addition, the peaks at 945 cm^−1^ result from the stretching vibration of the P-O-Ar structure, which was formed by the crosslinking reaction of BPS and HCCP. The FT-IR results indicate that the crosslinking reaction of BPS and HCCP occurred successfully and the outside part of SiO_2_ was PZS, which also shows that the three prepared core–shell structures obtained with the same preparation method possessed similar structures.

In order to further verify the successful coating reaction of BPS and HCCP, which can otherwise be understood as the so-called core–shell structure of the prepared SiO_2_@PZS nanoparticle, TEM analysis was performed (Figure 1). Compared to the SiO_2_ nanoparticles before coating, distinguishable core–shell structures could be observed in the fabricated SiO_2_@PZS nanoparticles from the TEM images, and the coated nanoparticles were uniform, which was very important for the subsequent membrane coating. Furthermore, by adjusting the reaction conditions, coatings with different shell thicknesses could be obtained, and three samples were fabricated with shell thicknesses of 65 ± 6 nm, 96 ± 6 nm and 128 ± 7 nm, respectively (Table 1). The diameters of the SiO_2_, SiO_2_@PZS-1, SiO_2_@PZS-2 and SiO_2_@PZS-3 particles were 281 nm, 411 nm, 473 nm and 537 nm, respectively. FT-IR and TEM results confirmed that a SiO_2_@PZS core–shell nanoparticle with uniform structure and controllable size was successfully prepared.

### 3.2. Characterization of Membranes for LIBs

The fabricated SiO_2_@PZS organic–inorganic composite was designed to improve the stability and electrochemical performance of PE membranes for use in LIBs. Thus, SiO_2_@PZS and a PVA binder were used to prepare the coating slurry, and the commercial microporous PE membrane was coated with the slurry on both sides. SEM measurements were first used to examine the surface morphology of the initial PE membrane and the PE-SiO_2_ and PE-SiO_2_@PZS composite separators (Figure 3). Compared to the initial PE membrane with a porous structure (Figure 3a), the PE-SiO_2_ (Figure 3b) and PE-SiO_2_@PZS (Figure 3c–e) nanoparticles were uniformly stacked on the surface of the PE membrane, and a porous structure was formed between the nanoparticles on the surface. The highly interconnected gap between the nanoparticles provided a good porous structure, which was expected to be filled with the liquid electrolyte, providing convenient channels for ion movement and improving the wettability of the membrane. Moreover, there was no obvious difference in the surface morphologies for the PE-SiO_2_@PZS-coated membrane with different PZS shell thicknesses. In addition, XRD tests showed that the PE-SiO_2_ and PE-SiO_2_@PZS-1 composite membranes were both amorphous (Appendix A).

As aspects of the physical performance (such as the electrolyte uptake and Gurley value) of separators play a pivotal role in LIBs, we then investigated those parameters in detail and the results are listed in Table 2. In order to facilitate the comparison, all of the separators were prepared with similar thicknesses. The Gurley value is a parameter that can be used to predict porosity and air permeability properties; a lower Gurley value indicates higher porosity and air permeability. We can conclude from Table 2 that the Gurley value of the separator coated with SiO_2_@PZS was comparable to that of the initial PE, which indicates that the coating process had a negligible effect on air permeability.

The electrolyte uptake of a separator is directly related to its electrochemical performance, and the cycle performance and conductivity of LIBs can be enhanced by improving the electrolyte uptake of the separators. The electrolyte uptakes of the separators coated with SiO_2_ and SiO_2_@PZS were both higher than that of the initial PE (Table 2). Interestingly, compared with the SiO_2_ coating, the electrolyte uptakes of the three SiO_2_@PZS coating separators were higher and the electrolyte uptake value increased with the increasing size of the PZS shell layer. The electrolyte uptake of PE-SiO_2_@PZS-3 increased by up to 213%. The increase in the electrolyte uptake for the nanoparticle coating separators mainly resulted from the pores between SiO_2_ and SiO_2_@PZS nanoparticles. Moreover, the crosslinked PZS shell layer of the SiO_2_@PZS also interacted with the polar organic solvents and lithium salt, which may explain the higher electrolyte uptake of the SiO_2_@PZS-coated separator compared to the SiO_2_-coated separator.

### 3.3. Characterization of Electrochemical Performance

Finally, we focused on the electrochemical performance of the coated separators. Figure 4a shows the AC impedance spectra for the simulated battery with a PE membrane and coated separators. The ionic conductivity (σ) calculated according to σ = L/(SR) is shown in Table 3. The σ value with the PE membrane was the lowest (5.8 × 10^−4^ S/cm) and that of PE-SiO_2_@PZS-3 was 1.4 × 10^−3^ S/cm, which was the highest. The σ values of the PE-SiO_2_@PZS-1-, PE-SiO_2_@PZS-2- and PE-SiO_2_@PZS-3-coated separators increased with the increase in the PZS shell thickness, and they were also higher than that of the PE-SiO_2_-coated separator, which was mainly due to the improvement in the electrolyte uptake provided by the PZS outer layer.

Figure 5 shows the rate performances of LiCoO_2_/Li batteries assembled with different separators. The PE membrane and composite separators were discharged at 0.5 C, 1 C, 2 C, 5 C and 8 C, respectively, under a charging rate of 0.5 C. It can be seen from Figure 5 that the discharge capacities of the composite separators were higher than for the PE membrane, and the discharge capacity decreased with the increase in the rate. The active substance adsorbed on the electrode decreased with the increase in the discharge rate, resulting in a lower discharge capacity. At the same rate, the discharge capacity of the PE-SiO_2_@PZS composite separator gradually increased with the increase in the thickness of the PZS outer layer. Moreover, the discharge capacities of the PE-SiO_2_@PZS-3 separator at different rates were higher than these of the PE-SiO_2_@PZS-1, PE-SiO_2_@PZS-2 and PE-SiO_2_ separators: the discharge capacities at 1 C, 5 C and 8 C were 154, 142 and 129 mAh/g, respectively, which was mainly due to the excellent wettability and retention of the PE-SiO_2_@PZS-3 separator in the electrolyte.

The impedances of the batteries after C-rate measurement were also tested (Figure 4b). The lower resistance for the PE-SiO2@PZS separators suggested better compatibility between the electrolyte-soaked separator and the lithium electrode. The cell assembled with the PE-SiO_2_@PZS-3 separator showed the lowest resistance, indicating a negative correlation between the thickness of the PZS outer layer and the battery resistance. The hydroxyl groups and N and O atoms on the surface of the PZS can coordinate with lithium ions to enhance the dissociation of lithium salts, thereby improving ionic conductivity and discharge capacity [36,37].

## 4. Conclusions

A series of core–shell structured silica-polyphosphazene nanoparticles (SiO_2_@PZS) with different PZS layer thicknesses were synthesized and coated onto both sides of a polyethylene (PE) microporous membrane to prepare composite membranes. The effect of the SiO_2_@PZS size and the PZS layer thickness on the electrolyte uptake, Gurley value and electrochemical performance of the separators was carefully studied. Compared with the pure silica-coated membrane and PE membrane, the PE-SiO_2_@PZS composite membrane had higher ionic conductivity. With the increase in the SiO_2_@PZS shell thickness, the electrolyte uptake and ionic conductivity gradually increased. The discharge capacity of the PE-SiO_2_@PZS composite membrane at 8 C rate was 129 mAh/g, which was higher than the values of 107 mAh/g for the PE-SiO_2_ composite membrane and 104 mAh/g for the PE membrane.

## Figures and Tables

**Figure 1 materials-15-04875-f001:**
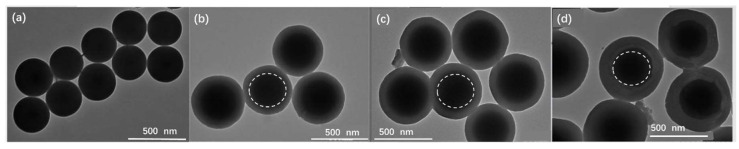
TEM images of SiO_2_ and coated core–shell nanoparticles: (**a**) SiO_2_, (**b**) SiO_2_@PZS-1, (**c**) SiO_2_@PZS-2, (**d**) SiO_2_@PZS-3.

**Figure 2 materials-15-04875-f002:**
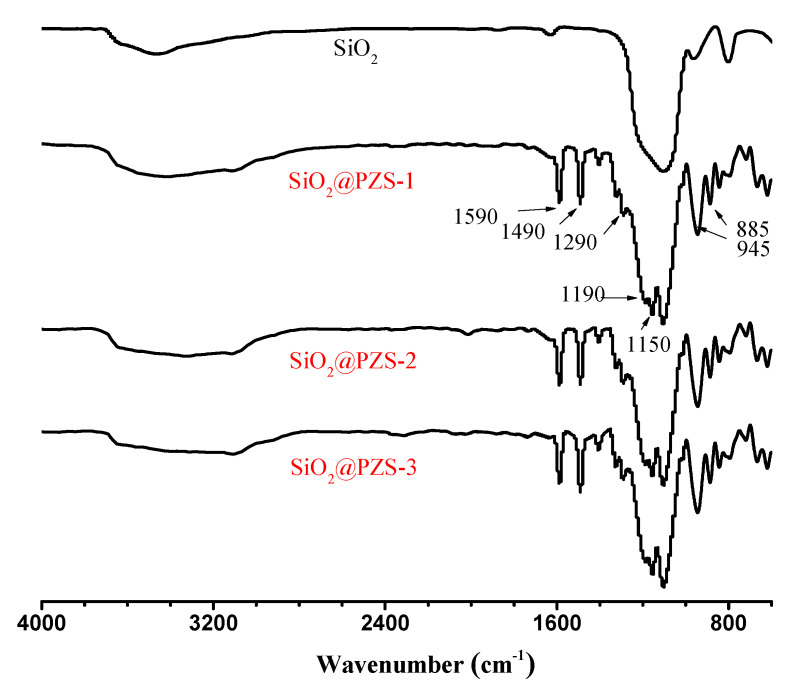
FT-IR spectra of SiO_2_, SiO_2_@PZS-1, SiO_2_@PZS-2 and SiO_2_@PZS-3.

**Figure 3 materials-15-04875-f003:**
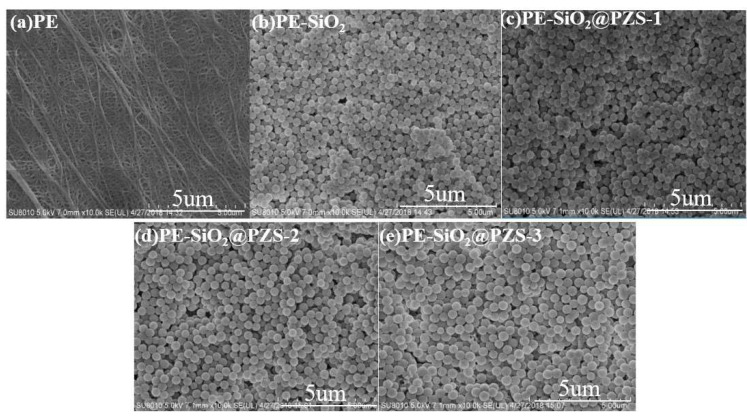
Surface morphologies of (**a**) PE membrane, (**b**) PE-SiO_2_, (**c**) PE-SiO_2_@PZS-1, (**d**) PE-SiO_2_@PZS-2 and (**e**) PE-SiO_2_@PZS-3 separators.

**Figure 4 materials-15-04875-f004:**
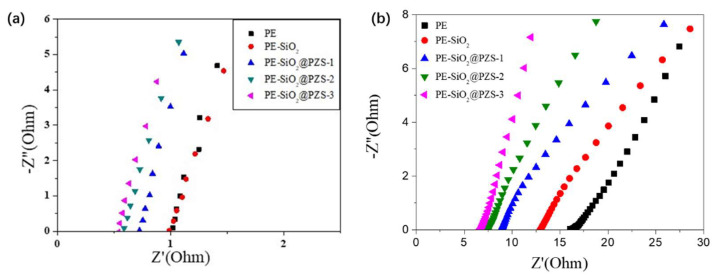
(**a**) The AC impedance spectrum responses of a stainless steel/separator/stainless steel sandwich structure for the PE membrane, PE-SiO_2_, PE-SiO_2_@PZS-1, PE-SiO_2_@PZS-2 and PE-SiO_2_@PZS-3 separators at room temperature. (**b**) AC impedance spectra for LiCoO_2_/Li batteries assembled with PE membrane, PE-SiO_2_, PE-SiO_2_@PZS-1, PE-SiO_2_@PZS-2 and PE-SiO_2_@PZS-3 separators after C-rate measurement.

**Figure 5 materials-15-04875-f005:**
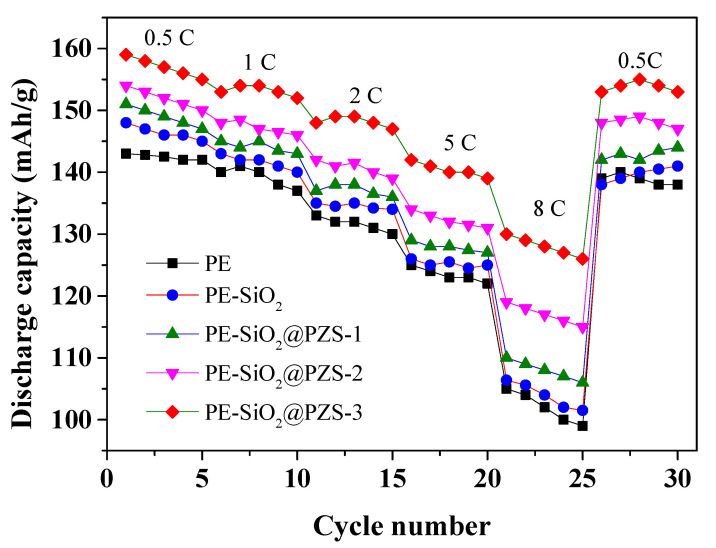
Discharge capacities of the cells assembled with the PE membrane, PE-SiO_2_, PE-SiO_2_@PZS-1, PE-SiO_2_@PZS-2 and PE-SiO_2_@PZS-3 separators at different discharge rates.

**Table 1 materials-15-04875-t001:** Synthesis of silica-polyphosphazene nanoparticles (SiO_2_@PZS).

Sample	SiO_2_ (g)	BPS (mol/L)	HCCP (mol/L)	CAN (mL)	TEA (mL)	Layer Thickness of PZS (nm) ^a^
SiO_2_@PZS-1	0.5	0.03	0.01	100	2	65 ± 6
SiO_2_@PZS-2	0.5	0.06	0.02	100	4	96 ± 6
SiO_2_@PZS-3	0.5	0.12	0.04	100	8	128 ± 7

^a^ Layer thickness of PZS was calculated with image software based on TEM analysis.

**Table 2 materials-15-04875-t002:** Physical properties of PE membrane, PE-SiO_2_, PE-SiO2@PZS-1, PE-SiO2@PZS-2 and PE-SiO2@PZS-3 separators.

Separator	Thickness (μm)	Gurley Value (s/100 mL)	Electrolyte Uptake (%)/1 h
PE	18	203 ± 1.5	90.0 ± 2.2
PE-SiO_2_	20.7 ± 0.5	215 ± 5.0	123.0 ± 9.0
PE-SiO_2_@PZS-1	21.8 ± 1.3	206 ± 7.2	138.6 ± 11.6
PE-SiO_2_@PZS-2	21.2 ± 1.3	204 ± 2.5	182.0 ± 14.5
PE-SiO_2_@PZS-3	22.3 ± 0.9	210 ± 3.5	213.5 ± 16.9

**Table 3 materials-15-04875-t003:** R_b_ and ionic conductivity of the PE membrane, PE-SiO_2_, PE-SiO_2_@PZS-1, PE-SiO_2_@PZS-2 and PE-SiO_2_@PZS-3 separators.

Separator	R_b_ (Ω)	Ionic Conductivity (S/cm)
PE	1.00	5.8 × 10^−4^
PE-SiO_2_	0.98	6.8 × 10^−4^
PE-SiO_2_@PZS-1	0.71	9.9 × 10^−4^
PE-SiO_2_@PZS-2	0.58	1.2 × 10^−3^
PE-SiO_2_@PZS-3	0.53	1.4 × 10^−3^

## Data Availability

Data sharing is not applicable for this article.

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
