# Peer review of "Fabrication and Investigation of PE-SiO2@PZS Composite Separator for Lithium-Ion Batteries"

_materials, 2022, doi:10.3390/ma15144875_

Round 1

Reviewer 1 Report

Title:

Fabrication and Investigation of PE-SiO2@PZS Composite Separator for Lithium-Ion Batteries

Comments:

The authors reported a composite membrane separator of PE-SiO2@PZS with different PZS layer thickness to lithium-ion battery applications. The searching for new separators with different properties to increase the battery performance is important to the energy field. Although that, the work present some drawbacks that should be clarified.

1-     Abbreviations as PVA, for example, should be explained at the first use. Please revise all the abbreviations at the manuscript.

2-     What is the active mass loading used at the cathode?

3-     How the SiO2@PZS was added to the PE membrane surface. It was controlled the thickness of the slurry in PE surface?

4-     PVA/PE membrane was prepared and their results compared? The pristine PVA influence at the surface should be also studied.

5-     What is the size of the synthetized particles? The overall sizes of the SiO2, SiO2@PZS-1, SiO2@PZS-2 and SiO2@PZS-3 particles should be mentioned.

6-     FT-IR were performed to all the prepared sample? The three prepared core-shell should be studied to verify if their preparation method has influence on the structure results.

7-     Why the charge rate is 0.5C to all the different discharge rates (0.5C to 8C)?

8-     More explanation should be provided to the electrochemical results. For example the discharge capacity decreasing with the discharge rate increasing.

9-     The EIS technique should also increase the understanding of the obtained results. To increase the manuscript quality the reviewer suggest the impedance of the batteries before and after cycling.

According to that, this referee believes that this work should considered to major revision.

Reviewer 2 Report

In this submitted manuscript entitled "Fabrication and Investigation of PE-SiO2@PZS Composite Separator for Lithium-Ion Batteries" authors tried to show the application of their composite material as a Lithium-Ion Battery studies. The approach is fine and very good but the provided studies are a primary phase of the Battery studies. Authors need to add more data’s regarding the materials well characterizations also in battery study. Below is the detail where authors need to add more characterizations and data’s to improve their manuscript.

For material characterizations

1.   Authors need to add the X-ray diffraction pattern for to show the size and phases of the material. Although SiO2 show amorphous character in nature but it depend their experimental conditions.

2.   For the size of the material in liquid medium authors need to add the Hydrodynamic size and zeta potential.

3.   Brunauer–Emmett–Teller (BET) specific surface area for their material.

4.   Due to authors studied the surface properties of the SiO2 and prepared a PE-SiO2@PZS Composite so it would be better for the readers to shows the X-ray photoelectron studies for their material.

5.   Atomic Force microscopy is also a good option to show the surface properties.

For Electrochemical Characterization

1.   Authors need to add the experimental relationship between the mass ratio and energy density. Also need to shows the galvanostatic charge–discharge curves. The electrochemical studies are also very limited. Need to add more information’s, although authors have not studied the super capacitors they have only limited study for battery. But, if possible need to add more detailed information’s and to make their manuscript valuable for the reader.

2.   Authors need to add the mechanisms and to show the relation to their material (composite PE-SiO2@PZS Composite) with battery application. Authors can take help from the previous published literature.

Reviewer 3 Report

Ms. Ref. No.: materials-1724988

The manuscript entitled "Fabrication and Investigation of PE-SiO2@PZS Composite Sepa- rator for Lithium-Ion Batteries " was studied.

The prepared separator for LIB is interesting. The synthesis and characterization sections were explained well and are acceptable. However, the electrochemical section is weak.

It needs a minor revision regarding following comments:

1.    In case of value of variables in Table 1, based on what criteria the mentioned values were chosen and examined.

2.    The optimization of the variables, including the thickness of PZS should be described.

3.    Is the 30 Cycles test enough for evaluating the cycle stability?

4.    The performance of developed separator in present work with previously developed ones should be compared in a table.

Round 2

Reviewer 2 Report

The submitted manuscript is well edited by the authors and now it is publishible in this journal as it is.